# Recent Progress and Perspectives on Neural Chip Platforms Integrating PDMS-Based Microfluidic Devices and Microelectrode Arrays

**DOI:** 10.3390/mi14040709

**Published:** 2023-03-23

**Authors:** Shihong Xu, Yaoyao Liu, Yan Yang, Kui Zhang, Wei Liang, Zhaojie Xu, Yirong Wu, Jinping Luo, Chengyu Zhuang, Xinxia Cai

**Affiliations:** 1State Key Laboratory of Transducer Technology, Aerospace Information Research Institute, Chinese Academy of Sciences, Beijing 100190, China; 2School of Electronic, Electrical and Communication Engineering, University of Chinese Academy of Sciences, Beijing 100049, China; 3Department of Orthopaedics, Rujing Hospital, Shanghai Jiao Tong University School of Medicine, Shanghai 200025, China

**Keywords:** microelectrode array, microfluidics, neural applications, nanomaterials, micro-electro-mechanical system

## Abstract

Recent years have witnessed a spurt of progress in the application of the encoding and decoding of neural activities to drug screening, diseases diagnosis, and brain–computer interactions. To overcome the constraints of the complexity of the brain and the ethical considerations of in vivo research, neural chip platforms integrating microfluidic devices and microelectrode arrays have been raised, which can not only customize growth paths for neurons in vitro but also monitor and modulate the specialized neural networks grown on chips. Therefore, this article reviews the developmental history of chip platforms integrating microfluidic devices and microelectrode arrays. First, we review the design and application of advanced microelectrode arrays and microfluidic devices. After, we introduce the fabrication process of neural chip platforms. Finally, we highlight the recent progress on this type of chip platform as a research tool in the field of brain science and neuroscience, focusing on neuropharmacology, neurological diseases, and simplified brain models. This is a detailed and comprehensive review of neural chip platforms. This work aims to fulfill the following three goals: (1) summarize the latest design patterns and fabrication schemes of such platforms, providing a reference for the development of other new platforms; (2) generalize several important applications of chip platforms in the field of neurology, which will attract the attention of scientists in the field; and (3) propose the developmental direction of neural chip platforms integrating microfluidic devices and microelectrode arrays.

## 1. Introduction

The brain is the most complex organ in human beings and the most sophisticated structure of nature. It is composed of hundreds of billions of neurons and their interconnected networks [1]. The complexity of the human brain can even be comparable to that of galaxies in the universe. Vazza et al. found that, although the formation process of the human brain neural network is completely different from that of the Milky Way cosmic network in the universe, they have formed a very similar organizational structure [2]. It is precisely because of the complexity of the brain that human beings have advanced functions, such as thinking, learning, perception, and decision making. Many diseases, such as epilepsy [3], Parkinson’s disease [4], and Alzheimer’s disease [5], are also closely related to the complexity of neural networks. To reveal the potential mechanism of the complex brain, scientists from different disciplines have made continuous efforts to develop and propose a neural chip platform [6] or brain-on-chip [7] for the research of major brain topics such as neuropharmacology, the diagnosis of nervous system diseases, and next-generation artificial intelligence [8]. Among the technologies involved in the neural chip platform, the microelectrode array (MEA) and microfluidic devices play pivotal roles. In particular, the MEA [9] bears the responsibility of recording neural information, while microfluidic devices play prominent roles in customizing the topology of in vitro neural networks.

The development of in vitro MEAs aims to explore the potential mechanism of production, processing, and transmission of neural information in the brain [10]. In vitro MEAs have been used by scientists from different disciplines for drug screening [11,12], biosensors [13,14], and pathological mechanism research [15,16] because of their noninvasiveness, high biocompatibility, high controllability, and the ability to read and modulate signals of neural cultures bidirectionally [17]. In the next generation of artificial intelligence that combines artificial intelligence with the brain, a neural chip platform with in vitro MEAs as the core component also shines brightly. The neurons cultured on an MEA, as an intelligent controller, successfully control the traveling of the mechanical trolley [18]. Furthermore, the Central Labs Company successfully taught the neurons on an MEA to play the classic video game Pong by electrical stimulation [19].

Although the neural chip platform containing only in vitro MEAs can cultivate 2D [20,21] or 3D [22] neural networks, the connections formed by neuron cultures in vitro are mostly complex and random, which is quite different from the brain network with specific functional connections. In addition, biochemical conditions in different regions of the brain are also different, and in vitro MEAs do not have the ability to provide a differentiated biochemical environment for local neural networks. Therefore, the neural chip platform continues to introduce a technology that can modularize and control the growth of neurons to build an in vitro model more suitable for the brain and use it in related research. Microfluidics is a technology to precisely control and manipulate microscale fluids, especially submicron structures [23,24]. We can use these characteristics of microfluidic technology to manufacture various microstructural units at the micron to submicron level, such as fluid channels [25], control channels [26], etc. These microstructures can modularize the neural network cultured in vitro while maintaining customized connections between modules [27,28]. It can be seen that the neural chip platform combining in vitro MEAs and microfluidic devices is not only a simplified platform in vitro for the brain but also a platform for neural information modulation and detection, which is a crucial tool in the research of nerves and their interdisciplinary subjects.

The purpose of this review is to discuss the latest trends in neural chip platforms integrating microfluidic devices and microelectrode arrays. For our purpose, we divided this article into three parts: First, the latest research progress on MEAs and microfluidic devices in the field of neural science is introduced, and the design and fabrication of microelectrode arrays and microfluidic devices are introduced. Next, we describe the latest developments on microfluidic and microelectrode array neural chip platforms from three popular application fields, namely, neuropharmacology, neural diseases, and simplified brain models. Finally, we put forward the development direction of the platform. A schematic diagram of this overview is shown in Figure 1.

## 2. An Important Tool for the Modulation and Recording of Neural Information: Microelectrode Arrays

### 2.1. In Vitro MEAs

Generally, microelectrodes, with diameters ranging from several microns to tens of microns, are patterned and arranged onto flexible or rigid substrates to produce MEAs by semiconductor micromachining technology. The MEA has both neural recording and modulation functions, among which neural modulation refers to the electrical stimulation of neuron cells through microelectrodes. Due to the small-sized feature of microelectrodes, MEAs can detect neural information with a high time–space resolution. The MEA can not only detect the local field potential (LFP) generated by multiple neuron activities but can also accurately record the action potential of a single neuron [30,31]. Therefore, MEAs have been used by scientists as an important research tool to further understand the spatial and temporal dynamics of the brain.

MEAs can be divided into implantable MEAs [32] and in vitro MEAs [33] according to detection objects. The detection object of implantable MEAs is generally the mammalian brain, while the detection objects of in vitro MEAs are biological samples, such as cells and brain slices. Compared with implantable MEAs, in vitro MEAs have the advantages of noninvasiveness and fewer ethical constraints. In vitro-cultured simple and controllable neural networks are more suitable for research concerning neural disease mechanisms [34], neural drug screening [35], new-generation brain–computer interfaces [36], etc., compared with complex networks in the brain. In addition, the in vitro microelectrode array is also an important part of the neural chip platform, the function of which is to detect and modulate changes in neuronal activities [37].

### 2.2. Methods to Improve the Performance of MEAs

The traditional structure of MEAs consists of a base layer, a conductive layer, and an insulating layer. The base layer is used to hold conductive and insulating materials. The metal layer includes electrode sites, a connecting wire, and contact sites for an external circuit for recording and transmitting neural activity. The insulating layer is used to protect the conductive structure that does not contact with neural cells.

The MEA detects the electrophysiological activity of neural cells through the electrode interface in contact with neural cells. The ideal electrode should have the ability of high time–space resolution detection due to the low impedance, small phase delay, and a high signal-to-noise ratio as well as the ability of precise and safe electrical stimulation due to a high charge injection density and a high maximum charge safe injection limit. These properties depend on the material and structure of the conductive layer. However, Pt, Au, ITO, TiN [38], and other materials commonly used as conductive layers in microelectrode arrays have some shortcomings in the performance of neural information recording and electrical stimulation. Therefore, an increasing number of scientists are focused on improving the performance of microelectrodes. In the current research, there are two main ways to improve the performance of microelectrodes. One research direction is to tightly cover the film with nanomaterials of excellent performance on the microelectrode by means of chemical modification to gain a better neural interface with high performance. The other is to improve the fabrication process of MEAs so that the microelectrode has a three-dimensional (3D) structure, which greatly improves the performance of microelectrodes.

Over the past decades, nanoscale metal particles and carbon nanomaterials have been widely used in the research of electrode modification. As shown in Figure 2A, platinum black, with a rough surface, is often deposited on MEAs to reduce the impedance of the electrode [32]. Nanoscale metal films can significantly increase the active surface area of microelectrodes thus improving the electrical performance of electrodes. However, there are still some deficiencies of nanoscale metal films in some research. For example, Park et al. found that the porous structure of platinum black is very fragile and easily damaged by external stimuli [39]. Tang et al. found that platinum nanoparticles lacked adhesion strength and durability in long-term experiments [40]. Strickland et al. determined that TiO_2_ particles have a certain neurotoxicity through cortical networks [41]. Carbon nanomaterials, represented by carbon nanotubes and graphene, have become good candidates for persistent neural interfaces due to their excellent conductivity, physical properties, and biocompatibility. Recent research shows that carbon-containing materials such as CNTs as neural interfaces have a low impedance, a high charge injection, and other characteristics and can also promote the attachment and growth of neurons (Figure 2B) [42,43]. Carbon materials such as graphene not only show a highly sensitive detection of neural electrical signals due to their high charge transfer ability [44] but also exhibit an excellent performance in electrochemical detection related to neurotransmitters and can detect the concentration of chemical transmitters such as dopamine at the nM level or even smaller [30]. In recent years, conductive polymers have become a new material in the field of neuronal recording and stimulation. Nanocomposites with higher performance can be formed by mixing two or more nanomaterials reasonably. Generally, composites can cover up the defects of single-component materials and maximize the advantages of each component material. Currently, nanocomposites made of poly (3,4-ethylenedioxythiophene) (PEDOT) or polypyrrole (PPy) and other materials have been used in MEAs in various scenarios [45,46]. Saunier et al. synthesized the nanocomposite material PEDOT/carbon nanofibers(CNF) using CNF with excellent mechanical stability and PEDOT with good biocompatibility and proved that the PEDOT/CNF film can promote the adsorption, growth, and division of neurons in vitro (Figure 2C) [47]. Xu et al. showed a new nanocomposite material carboxylated graphene(cGO)/PEDOT:poly(styrenesulfonate)(PSS) with significant impedance characteristics, charge injection ability, and a high active area, and they successfully activated the learning and memory function of hippocampal neurons in vitro using the MEA modified with cGO/PEDOT:PSS (Figure 2D) [31]. He et al. modified the highly sensitive nanocomposite material rGO/PEDOT: PSS on the MEA with four chambers and successfully detected the quantized release of dopamine from dopaminergic neurons induced by potassium ions [48].

Although the method of material modification can improve the performance of microelectrodes, it is difficult to meet some of the special needs of MEAs in neuroscience research (such as subthreshold potential measurement and drug delivery). By optimizing the preparation process, changing the morphology of microelectrodes can enable MEAs to add the required capabilities. Because 3D microelectrodes can enhance electrical coupling with neurons, thereby improving the performance of the MEA in neural recording, much innovative research has emerged in this field in recent years [49]. As shown in Figure 2E, Hai et al. modified the polypeptide on a mushroom-shaped gold electrode for recording intracellular signals. Because of the phagocytosis of the nerve cell membrane, the microelectrode can record the intracellular potential such as the patch clamp, which enables MEAs to reflect the changes in neural information more carefully [50,51,52]. Liu et al. further combined MEMS technology with nanowire preparation technology to develop nanowire microelectrode arrays that can simultaneously detect changes in the extracellular action, potential, and intracellular subthreshold potential of neurons (Figure 2F) [53]. In addition, Bruno et al. used ion beam milling technology to prepare an MEA with a 3D nanotube structure, which could conduct local drug delivery and chemical stimulation on the cultured neural culture (Figure 2G) [54]. In terms of simplifying the fabrication of 3D MEAs, Zips et al. skillfully produced 3D microelectrodes with conductive polymer ink printing and evaluated their performance in neural recording and electrical stimulation (Figure 2H) [55].

**Figure 2 micromachines-14-00709-f002:**
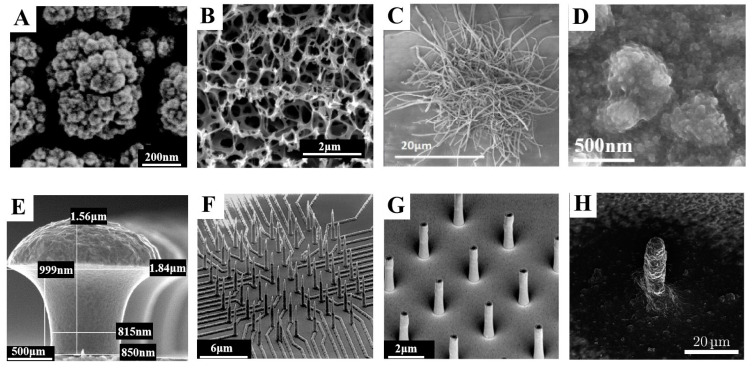
Methods for improving, recording, and stimulating properties of microelectrodes. (**A**) SEM diagram of platinum black [32]. Reproduced with permission from Xu, Microsystems & Nanoengineering; published by Nature, 2022. (**B**) SEM of surface-porous graphene [43]. Reproduced with permission from Lu, Scientific Reports; published by Nature, 2016. (**C**) SEM diagram of PEDOT: CNF [47]. Reproduced with permission from Saunier, Biosensors and Bioelectronics; published by Elsevier, 2020. (**D**) SEM diagram of cGO/PEDOT:PSS [31]. Reproduced with permission from Xu, Biosensors; published by MDPI, 2022. (**E**) SEM diagram of a 3D golden mushroom electrode [52]. Reproduced with permission from Hai, Journal of the Royal Society Interface; published by Royal Society, 2009. (**F**) SEM images of nanowire MEAs. Reprinted with permission from [53]. Copyright 2020 American Chemical Society. (**G**) SEM diagram of nanotube microelectrodes with local substance delivery functions [54]. Reproduced with permission from Bruno, Frontiers in Bioengineering and Biotechnology; published by Frontiers, 2020. (**H**) Structure of a 3D microelectrode printed with ink [55]. Reprinted with permission from. Copyright 2019 American Chemical Society.

### 2.3. Development Direction of In Vitro MEAs

The basic model of in vitro MEAs can be traced back to the 1970s. Thomas et al. created an MEA for detecting myocardial cells [56]. With the increasing demand for in vitro neural information detection, brain–computer interactions, biosensors, and the growing maturity of micromachining technology, in vitro MEAs have become one of the most important research tools in the field of neurology.

In recent years, in vitro MEAs as a research tool have been widely used and innovated in the following aspects. (1) According to the actual needs of researchers, it will develop towards customized MEAs. As shown in Figure 3A, an MEA was customized for the study of neural information in hippocampal slice circuits [57]. (2) MEAs are developing towards having a higher density and larger area, while the development of high-density MEAs needs to be rooted in CMOS technology, and there have been many research efforts in this regard (Figure 3B) [58,59,60]. (3) The combination of an MEA and other technologies forms a more effective in vitro detection platform. As shown in Figure 3C, the in vitro detection platform combining an MEA and microfluidic technology is a successful example [61]. MEAs may also be combined with μLED to become in vitro optical electrodes [62,63]. (4) The microfabrication technology and materials of MEAs are being optimized to achieve a lower cost and faster preparation. In addition, the use of SU-8 to replace traditional SiO_2_ and Si_3_N_4_ insulation layers [64] and 3D printing technology to easily make MEAs (Figure 3D) [65] are pioneering attempts in this field.

## 3. An Important Tool for Neural Network Customization: PDMS-Based Microfluidic Devices

Microfluidics is a new cross-discipline developed based on microelectronics, nanotechnology, fluid physics, chemistry, biology, etc. [66]. Microfluidic devices are the core of microfluidics, which are generally composed of microchannels, micropumps, microvalves, micromixers, and other components, and they can conduct the integrated processing of biological or chemical samples [67,68]. Because of their numerous applications in biology, microfluidic devices are also considered to be a bridge between life sciences and information science [69,70].

In this section, we first review the characteristics and applications of PDMS-based microfluidic devices. Then, we introduce the history and the frontiers of studies on a special and important PDMS-based microfluidic device which supports controllable neurite growth. Furthermore, the applications of the devices used in the field of neuroscience are reviewed, such as drug screening and the mechanical study of neurological disorders.

### 3.1. PDMS-Based Microfluidic Devices

Common materials used in microfluidic devices include silicon, glass, polymer, and paper. The microfluidic devices made of each material play an important role in the research of corresponding fields. In the research of cell cultures, polymer-based microfluidic devices have attracted much attention.

Since polymers were used in the field of microfluidics, they have always been the preferred materials for commercial applications and high-throughput systems. Polydimethylsiloxane (PDMS) has been the most widely used material in academic research because of its excellent properties, such as low cost, optical transparency, good biocompatibility, strong permeability, and good plasticity. PDMS-based microfluidic devices are often used in biochemical sample separation and cell culture-related research. For example, Jeon et al. designed a plastic spiral inertial microfluidic system for the high-throughput separation of blood and cells [71]. Huang et al. designed a microfluidic device for real-time and large-scale drug screening, which can screen 10 drugs at the same time [72]. Microfluidic devices of PDMS also have important applications in the neural field, which are described in detail later.

### 3.2. Development and Latest Design of PDMS Microfluidic Devices with Compartments of Controllable Neurite Growth

The application of microfluidic devices with isolation chambers in the field of neurology can be traced back to the research by Campenot in 1977 [73]. He divided a culture dish into three different chambers, inoculating neurons in the central chamber and adding different concentrations of growth factors in the lateral chamber to study the growth of synapses. Campenot’s innovative research enabled future researchers to conduct accurate physical processing and biochemical analysis of an isolated synaptic part. However, the improvements by Taylor and other scientists promoted the wide application of such devices in subsequent research. At the beginning of the 21st century, Taylor’s team designed and produced a neuron culture device, which allows for the isolation of neurites and cell bodies [74,75]. Figure 4A shows the structure of the device. Specifically, the device has two compartments that provide a biochemical environment for neurons. The different compartments are connected by several micron-level grooves, allowing neurites to grow in the middle while maintaining fluid isolation at both ends. While designing and developing the microfluidic device, Taylor et al. also used the microfluidic platform to construct an in vitro model of axon injury and found the characteristics of gene expression at the early stage of axon injury [76]. In a subsequent study, Taylor and his colleagues improved the microfluidic platform and developed a local perfusion chamber based on the original one to control the neurite region between neurons at both ends [77].

Many researchers have made innovations in the structural design of microfluidic devices according to research needs. In these studies, researchers can increase the number of shapes of the cull culture chambers according to research needs to achieve the research purpose. For example, Vitis’ [78] and Coquinco’s [79] teams developed devices with multiple chambers for the co-culturing of multiple cell populations. As shown in Figure 4B, Park’s team developed a microfluidic device with six axon chambers surrounding a cell compartment, which can separate the cell body and dendrites of neurons and guide axon growth [80]. In addition, Megartiy et al. [81] proposed a modular microfluidic platform that can be spliced, as shown in Figure 4C. The platform is composed of several independent microfluidic devices, which can be spliced similar to a puzzle through the protrusions and depressions. They also cultured hippocampal neurons on the platform to verify the maintenance characteristics of neurite connections at the module interface. This method provides an idea for users to freely assemble microfluidic platforms.

The advanced design of interconnecting microchannels is another direction of the innovative research on microfluidic devices. As shown in Figure 4D, Gladkov and his colleagues designed a variety of asymmetric microchannels to promote neurons to grow in one direction [82]. This study shows that a microchannel containing repeated triangles has a better ability to promote the unidirectional growth of neural processes. Renault et al. designed a microfluidic device with microchannels of “arch” in which unidirectional growth can be achieved by reintroducing neurites that do not require directional growth into the original compartment [83]. In addition to the above shapes, hook-shaped microchannels [84], barb-shaped microchannels [85], and microchannels from wide to narrow [86] have also been developed to promote the unidirectional growth of neurons growing in compartments. Based on previous research, Forró’s team developed a “stomach”-like microchannel, as shown in Figure 4E [28]. This shaped channel can greatly improve the efficiency of neurite growth between two nodes, and the success rate of unidirectional growth can reach 95%. Girardin et al. [87] and Ihle et al. [88] used microfluidic devices to study the response of user-defined neural networks to electrical stimulation, and the design of the microfluidic devices was inspired by Forró et al. [28].

**Figure 4 micromachines-14-00709-f004:**
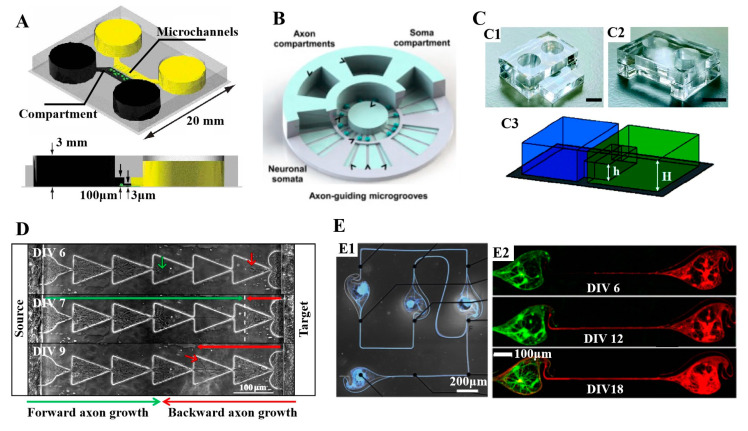
PDMS microfluidic devices with controlling synaptic growth channels. (**A**) Structure diagram of a microfluidic device capable of separating cell bodies and neurite [76]. Reproduced with permission from Taylor, Nature Methods; published by Nature, 2005. (**B**) A microfluidic device with a plurality of axon chambers surrounding the axon chambers [80]. Reproduced with permission from Park, Journal of Neuroscience Methods; published by Elsevier, 2014. (**C**) A modular platform consisting of multiple individual microfluidic units that can be combined in several configurations to create bespoke culture environments [81]. Reproduced with permission from Megarity, Lab on a Chip; published by Royal Society of Chemistry, 2022. (**D**) An asymmetric microchannel that controls the unidirectional growth of neurite [82] DIV 6, DIV 7, and DIV 9 represent neurons cultured for 6, 12, and 9 days, respectively. Reproduced with permission from Gladkov, Scientific Reports; published by Nature, 2017. (**E**) A “stomach”-shaped microchannel that controls the unidirectional growth of neurite [28] (DIV 6, DIV 12, and DIV 18 represent neurons cultured for 6, 12, and 18 days, respectively). Reproduced with permission from Forro, Biosensors and Bioelectronics; published by Elsevier, 2018.

### 3.3. Important Applications of Compartmental PDMS Microfluidic Devices with Controlled Neurite Growth PDMS in Neuroscience

Such microfluidic devices for neuronal cultures with controllable neurite growth have been important research tools in the field of neuroscience since their development and have led to important achievements in the research of the pathological mechanisms of neural diseases and simplified models of brain regions in vitro [89,90].

In the study of neurological diseases using microfluidic devices, the drug-induced pathological model can be constructed by controlling the concentration of substances in the chamber. Kunze et al. constructed a novel experimental model of Alzheimer’s disease in vitro [91]. Specifically, they added Okada acid to the compartment on one side of the microfluidic device which caused the hyperphosphorylation of the Tau protein of neurons in the compartment (one of the main signs of Alzheimer’s disease). This model can be used to study the transmission of cellular neural information on both sides of “health” and “disease”. In a similar study, Kajtez’s team aimed to use microfluidic devices to study the mechanism of neural information transmission in the pathogenesis of Parkinson’s disease in vitro [92]. They built a nigrostriatal pathway related to Parkinson’s disease by cultivating human-induced neurons in the connected chamber. Virlogeux et al. developed a cortical–stratal network on a chip by combining microfluidic devices with high-resolution microscopes [93]. In addition, microfluidic devices are also used in the research of epilepsy [94], inflammation [95], amyotrophic lateral sclerosis [96], brain tumors [97], and other disease models.

Advanced functions of the neural system are closely related to the connections between neural networks. Therefore, the co-culturing of different cell populations and the specific synaptic connections between neurons are two important and necessary conditions for establishing a simplified model that can simulate part of the brain’s function in vitro [98,99]. This kind of microfluidic device with different chambers can well meet these two conditions. Neuron–glial cells [100], neuron–muscle cells [101], neuron–cancer cells [102], neuron–Schwann cells [103], etc., were cultured into microfluidic devices by researchers to build models for researching the transmission of information among different cells. These models can be used to study the communications between cell populations related to the brain and play important roles in understanding the closed-loop feedback mechanism of the neural system and neural network dynamics. For example, Berdichevsky’s team developed an in vitro platform based on microfluidic devices to study neural pathways between different brain regions by co-culturing brain slices of different brain regions (such as the hippocampus and entorhinal cortex) in the device [104]. Berdichevsky applied this platform to study the connectivity of neuronal circuits, synaptic activity, and drug screening, respectively.

## 4. Fabrication of Neural Chips Integrating Microfluidics and Microelectrode Arrays

### 4.1. Fabrication of In Vitro MEAs

The manufacturing of in vitro MEAs mainly depends on two basic processes: MEMS and CMOS [105]. Commonly used planar and 3D MEAs are mostly prepared based on MEMS technology [106,107], while the preparation of high-density MEAs, requiring a higher spatial resolution, is based on a CMOS process [108]. Planar MEAs contain components such as recording sites, counter electrodes, contact sites, connecting wires, and external circuits. The manufacturing process of planar MEAs includes mask making, multiple lithography, multiple material deposition, etching, and surface treatment. These processes must be completed in a super clean room [31]. The basic process flow is shown in Figure 5A. Before making MEAs, the substrate is generally cleaned to remove pollutants from the surface and increase hydrophilicity. Next, the conductive film is patterned onto the substrate by means of photolithography and physical vapor deposition. Commonly used physical vapor deposition processes include vacuum evaporation and magnetron sputtering. The conductive layer materials are often Pt, Au, TiN, and other materials with good conductivity and biocompatibility. Sometimes, to better observe the growth of cells cultured on MEAs, transparent conductive materials such as ITO [109] and PEDOT [110] are also often used for conductive layers. After the conductive layer is deposited, it is necessary to add insulation protection to the parts, except to the recording sites, counter electrode, and contact sites. Generally, plasma-enhanced chemical vapor deposition is used to deposit insulating materials on the surface of MEAs as insulating layers, and the parts to be exposed are opened by etching technology [111].

So far, various 3D MEA preparation methods have been reported [112]. These preparation methods are mostly improved based on planar MEA. The typical method of manufacturing 3D MEAs is to use photolithography to make micropores, then deposit gold into the micropores through electrochemical deposition, and finally to remove the photoresist to obtain mushroom-shaped microelectrodes [113]. Nanoimprint stripping, electron beam lithography, and focused ion beam scanning techniques are also often used to make 3D nanoelectrode arrays. In addition, 3D printing technology is also a powerful tool for making high-resolution microstructures, which can realize complex 3D electrode microstructures at the microscale. These nano-microelectrodes can extract electrical and electrochemical information on synapses [114].

### 4.2. Fabrication of PDMS-Based Microfluidic Devices

The appearance of polymer microfluidic devices represented by PDMS has optimized the fabrication of microfluidic devices [115]. PDMS microfluidic chips can be produced quickly and cheaply in any standard laboratory through a relatively simple process flow. Figure 5B shows the standard method for copying and manufacturing PDMS microfluidic devices. The manufacturing process of this method is relatively simple, which can be divided into mold manufacturing and PDMS microfluidic device manufacturing according to the sequence of processes (Figure 5B). First, the mold is made by microprocesses (the specific method for making the mold is described in detail later). Then, the PDMS-mixed solution containing a certain proportion of crosslinking agent (usually the weight ratio of PDMS to curing agent is 10:1) is poured into the mold and baked at a high temperature to quickly cure and shape it [116]. Finally, after the PDMS is demolded, microfluidic devices with different microstructures can be manufactured in large quantities. The internal morphology, structure, size, and other characteristics of the microfluidic devices are mainly determined by the mold; therefore, how to make high-quality molds has become the key problem in the manufacturing process of PDMS microfluidic devices.

**Figure 5 micromachines-14-00709-f005:**
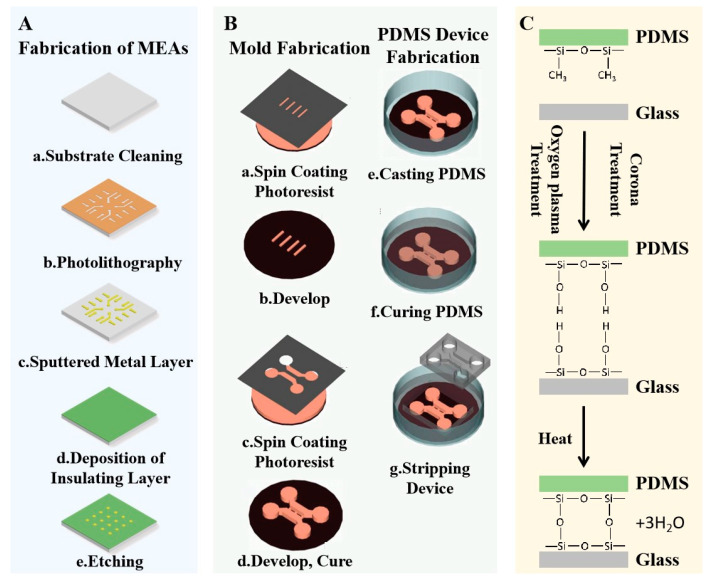
Fabrication process of a neural chip platform integrating microfluidics and MEAs. (**A**) Fabrication process of planar in vitro MEAs, including substrate cleaning (a), photolithography (b), sputtering a metal layer (c), deposition of an insulating layer (d) and etching an open window [31]. Reproduced with permission from Xu, Biosensor; published by MDPI, 2022. (**B**) Fabrication process of microfluidic devices based on SU-8 mold, including mold fabrication (a–d) and PDMS device fabrication (e–g) [116]. Reproduced with permission from Park, Nature Protocols; published by Nature, 2006. (**C**) Surface activation-based bonding method.

Commonly used PDMS microfluidic molds are still made on a substrate using the SU-8 photoresist [117,118]. This type of mold is made by patterning SU-8 on the substrate at the micron or submicron level through photolithography and has a high aspect ratio. Figure 5B shows the process of making an SU-8 mold. First, the SU-8 photoresist is spin coated on the substrate and then soft baked according to the thickness of the SU-8 photoresist. Next, the mold is exposed to ultraviolet light and baked to accelerate the SU-8′s polymerization so that the micropatterns on the mask are transferred to the SU-8 layer. The above photolithography steps are repeated for different types of SU-8 photoresists on the substrate to produce multilayer molds with different heights. Through these molds, microfluidic devices with complex structures can be copied.

Because mold manufacturing needs a certain amount of time and additional manufacturing costs, it has become another area of research interest to discard the replication program and directly build microstructures in PDMS. Recently, several methods for direct patterning on a PDMS surface by photolithography have been reported. Chen et al. reported a high-precision, repeatable micromachining method for PDMS surface patterning [119]. They spin coated the PDMS mixture on the activated substrate, then spin coated the photoresist on the PDMS layer as a protective layer, and finally etched the pattern on the PDMS surface through reactive ion etching. The work of Gao [120], Bhagat [121], and other teams proposed another method of surface graphics, which is to add photoinitiators to PDMS to produce photosensitive PDMS (photoPDMS) so that PDMS can be directly graphed. In addition, laser cutting technology is also an effective method to construct PDMS surface micrographics. Li et al. used laser ablation technology to build microchannels on PDMS and bonded PDMS devices with the substrate to form a closed microfluidic system for the study of dielectrophoretic [122]. Oyama’s team also used low-energy electron beam irradiation to build microchannels on the surface of PDMS devices [123].

### 4.3. Bonding of In Vitro MEAs and Microfluidic Devices

In general, PDMS microfluidic devices that build microchannels and chambers need to be closed through the substrate to form a closed chip [124,125]. In a neural chip platform composed of an MEA and microfluidic equipment, the MEA acts as the closed substrate of the microfluidic equipment, and the microelectrode of the MEA is located in the microfluidic channel or chamber to detect electrophysiological signals [126]. This method of closing microfluidic devices and substrates is called bonding because irreversible chemical bonds are formed between the substrates and the activated or functionalized surfaces of PDMS.

At present, the bonding methods commonly used for PDMS microfluidic devices include plasma surface activation [127], corona-treated surface bonding [128], ultraviolet/ozone-treated surface activation [129], chemical bonding [130], and adhesive bonding [131]. Plasma surface-activated bonding, corona-treated surface bonding, and ultraviolet/ozone-treated surface-activated bonding all belong to the category of surface activation and are the most common methods for sealing microchannels and chambers. Surface activation is mainly used for the bonding of PDMS and silicon-based materials (silicon or quartz glass). Its basic purpose is to remove surface contaminants and generate covalent reactive groups. Usually, the method of sealing by surface functionalization on the surface of the target substrate and PDMS is called chemical bonding, which is generally used in the bonding research between the PDMS and plastic substrate. Sometimes, in addition to functionalizing the surface, it is necessary to add adhesives to achieve good sealing between the PDMS and substrate, which is called adhesive bonding.

Because the substrate material of MEAs is silicon or glass, surface activation is commonly used in the bonding process between MEAs and PDMS microfluidic devices. Among the surface activation methods, studies have shown that the method of corona treatment will damage the metal film on the substrate [132], which is not suitable for bonding. Plasma surface activation, especially oxygen plasma surface activation, is the mainstream bonding method for MEAs as a substrate. Figure 5C shows the process of bonding MEAs with PDMS microfluidic devices. The prepared MEAs and microfluidic devices are put into a plasma cleaner for surface activation. After surface activation, the end methyl group is replaced by the silanol group. Then, the MEAs and microfluidic devices are quickly aligned on the alignment instrument and put onto a hot plate to remove the water molecules. After the loss of the water molecules, the PDMS forms covalent siloxane bonds with the silanol groups displayed on the surface of the MEAs, thus forming a seamless structure.

## 5. Neural Chip Platforms Integrating Microfluidic Devices and Microelectrode Arrays Play a Key Role in the Application of the Neural Field

### 5.1. Recording and Modulation of Neural Signals

The combination of an MEA and microfluidic devices can be used to monitor the generation and transmission of neural information within or between modular neural networks in the compartment. In the most relevant studies, microelectrodes in MEAs are either located in the microchannel or distributed at the inlet and outlet of the microchannel (Figure 6A). The microelectrodes distributed near the microchannel solve the problem of the difficult detection of axon electrical signals due to the small space size, complex connection, and difficult separation of the axons. Many researchers use this platform for information transmission between modular networks and axon signal research [133,134,135]. For example, Hong’s [136] and Goshi’s [137] teams found that the amplitude of the action potential of the axon in the microchannel was significantly higher than that of the cell body in the compartment, and they found that the microchannel would increase the noise of the microelectrode (Figure 6B). Hong et al. estimated the conduction velocity of the axons, which gradually becomes faster as the network matures. Goshi et al. further studied the influence of microchannels on microelectrode performance and electrophysiological record fidelity. The research shows that the increase in the noise of the microelectrode in the channel is largely attributed to the increase in electrode impedance, which may be caused by space limitation and cell blockage on both sides of the channel. To analyze the neural information in compartmentalized microfluidic devices in a more friendly manner, Heiney et al. developed an open and easy-to-use neural information computing tool uSpikeHunter [138]. This tool can well quantify the electrophysiological parameters of the axons in the microchannel, such as the rate of discharge, direction and speed of the signal propagation, and origin of the signal. Neural modulation is also an essential function of the MEA-based in vitro neural platform [139]. The activities of the neurons or neural networks can be promoted or inhibited by electrical, optical, or chemical means. Studies have shown that electrical stimulation is effective in studying synaptic plasticity [140], axonal regeneration [141], network reorganization [139], etc. Because the potential ability of electrical stimulation technology in neurite research meets the demand of neural platforms combining MEAs and microfluidic devices to study neurite activity locally, it is widely used. For example, Kim et al. found that electrical stimulation or neurotrophic factors can not only promote the growth of axons but also enable neural stem cells to further differentiate into neural cells by using such neural platforms [141]. As an important auxiliary tool, optical microscope imaging technology is used to characterize the state of axons in cell bodies and microchannels in the chamber. After the specific immunofluorescence staining of nerve cells, observation under a confocal microscope can provide researchers with a large amount of internal information on the microstructure, such as the distribution of the various nerve cells and the growth and connection of axons in the microchannels. (Figure 6C) [29,137,142]. In addition, Moutaux et al. used calcium imaging technology and MEA to study the electrical activity and calcium kinetics of synapses in the compartment under electrical stimulation and local drug modulation (Figure 6D) [143].

### 5.2. Neuropharmacology Research

Toxicity research and efficacy testing of new drugs are crucial in drug development. These tests can ensure the integrity of drug safety and efficacy. Traditional models for drug development and toxicological research include in vivo and in vitro models. In vitro cell models have become commonly used objects for drug screening research because of their low cost and absence of ethical issues. At the same time, nerve cells are important research objects of pharmacological research related to neurological diseases. MEAs can reflect the influence of drugs on nerve cells by recording the changes in the electrophysiological activities of neurons before and after exposure to chemicals. Because of this ability, MEAs have become an important research tool in drug screening and neurotoxicity detection [144]. In drug development and toxicology research related to cell models in vitro, microfluidic devices are also one of the most popular research tools. The superiority of microfluidic technology lies in its high throughput analysis capability [145].

Microfluidic devices with multiple chambers and multiple microchannels are often used for the chemical stimulation of local cells and can be used for dynamic concentration gradient drug reaction and high-concept drug screening. A microfluidic device combined with MEA technology can directly reflect the dynamic change process of drug screening through electrophysiological signals. Kraus’s team integrated MEAs with microfluidic devices and detected changes in the electrophysiological information of myocardial cell cultures during local drug stimulation [146]. Biffi et al. developed an MEA microfluidic device platform for the local chemical stimulation of neural networks [147]. In this platform, the culture area of neural cells was divided into two compartments, and the neural networks in each compartment were cultured in the same biochemical environment. The network was stimulated by injecting tetrodotoxin into the compartment separately to evaluate the neurotoxicity of tetrodotoxin. The research allowed for the controlled delivery of substances to local cells, overcoming the limitations of traditional drug screening research. Because of this local drug delivery capability, neural chip platforms integrating microfluidics and electrode arrays have also been used in many studies that require local targeted pharmacological operations [148,149], such as targeted drug therapy for neuritis [142].

### 5.3. Research of Neurological Diseases

Nervous system diseases affect hundreds of millions of people all over the world, and this continues to increase every year [150]. The use of in vitro neural cell models has always been an important tool for the study of nervous system functions and nervous system diseases. At present, a neural chip platform combined with microfluidic devices and MEAs can overcome the defect of the poor complexity of neural network structures and functions of previous cell models and can reflect neural information under normal or abnormal neural network conditions in real time through MEAs. Since microfluidic devices with chambers and microchannels were created to solve the problem of neurite from the beginning, research on neural diseases related to neurite injury or nerve regeneration is a direction of further research for this kind of neural chip platform. Wijdegen’s team developed a microfluidic platform integrating an MEA in a three-compartment series to study the plasticity of synaptic connections after being cut [151]. Wijdegen’s team borrowed the method of Tong et al. [152] to eliminate the neurite in the connecting channel. Specifically, they used a pipette to introduce gas into the channel, where it generated bubbles and cut off most of the axons. Figure 7A shows the growth of axons in microchannels before and after axotomy. Finally, they used an MEA to study the dynamic response of the neural networks during resection. In addition to physical methods, laser cutting [153] and chemical drugs [103] are also common methods for modeling axonal injury. Generally, after damage modeling on this kind of neural chip platform, researchers will also study the dynamic electrophysiological process of nerve regeneration induced by drugs or electric pulses [141].

In in vitro studies of neurodegenerative diseases, such as Alzheimer’s disease, Parkinson’s disease, and epilepsy, neural chip platforms with integrated microelectrode arrays and microfluidic devices also play an important role [150]. As shown in Figure 7(B1), Pelkonen et al. established an in vitro epileptic modular platform using MEA–PDMS chips to study the epileptic activities of neural networks [154]. They designed a closed-loop three-zone chamber microfluidic device and provided a gas supply module and a sealed plastic cover. As shown in Figure 7(B2), kainic acid (KA) was added to a compartment to induce the neural network to produce epileptic-like electrical activity, and the process of epileptic signal transmission was reflected by studying the changes in the activity of neurons in the connecting compartment. In addition, they also used the anticonvulsant phenytoin to study the effect of treating epileptiform discharges on neurons in different regions.

### 5.4. The Study of the Brain by Means of Simplified Model

The brain is the most complex structure in nature. This complexity comes from the complex structure of single neurons and the interlaced connection of billions of neurons. To study how the nervous system generates firing rhythms, encodes external stimuli, and reacts to drugs, researchers have been developing simplified experimental models of the extracorporeal brain in recent decades. The simplified brain model is also referred to as the brain-on-chip in many studies [98]. To build an ideal brain-on-chip model, we must first understand the basic characteristics of the brain. The function of the brain is closely related to the interactions among different regions, and each region has a unique structure and function; for example, the hippocampus has a specific dentate gyrus structure. From this, we can infer that building an ideal in vitro brain model should at least have the characteristics of modular connection topology and heterogeneity of neuron types (Figure 8A).

Modularization is the key feature of the brain model in vitro, and the chip platform integrated with microfluidics and microelectrode arrays has solved the problem of the modular cultivation and detection of neurons. To reveal the relationship between the structure and function of modular neural networks, Park et al. constructed three types of modular cortical networks and studied the connection between the neural activities and modules [27]. As shown in Figure 8B,C, they inoculated cortical neurons into four, three, and two culture chambers and recorded them as M4, M3, and M2. By estimating the size of the axonal bundles in the stained channels, it was determined that the modular connection strength decreased according to M4, M3, and M2. In addition, from the recorded spike signal, it can be found that when the connections between modular neuronal networks are strong, the neural signal transmits symmetrically in most microchannels, while when the connections between the networks are not strong, the direction of neural information transmission is asymmetric. According to the known connections of brain regions or subregions, the model designed to study the connections between neuronal networks is more suitable for studying the brain. Poli’s team [155] and Vakilna’s team [156] used an MEA integrated with a dual compartment microfluidic device as a tool to build a simplified model for studying the information transmission between various subregions in the hippocampus circuit in vitro. In the work of Poli et al. [155], they inoculated neurons with the same area density ratio as each subregion in the brain in the microfluidic device and stimulated and recorded the information transmission characteristics between regions through microelectrodes. They cultivated hippocampal slices on this chip platform to study the neural circuit of neural information transmission in the hippocampus. It was found that the information was projected from the dentate gyrus (DG) area to the cornu ammonis 3 (CA3) area and then to the cornu ammonis 1(CA1) area.

Compared with a single type of neural cell, the neural chip platform integrating microelectrode arrays and microfluidics is used more for the co-culture of heterogeneous cells (Figure 8A). In this way, researchers try to build connection models between different functional brain regions in vitro to better reveal the mysteries of brain science. Hippocampus–cortex [157], cortex–striatum [93], cortex–thalamus [158], forebrain–midbrain neurons [159], and other brain regions have been used in the study of simplified brain models. Recently, Chang’s team studied the hippocampus–neocortex co-cultured brain model from embryonic rats (Figure 8D) and found that this model can not only retain the original rhythmic activities of a single neural network but can also complete the communication among heterogeneous networks composed of different neurons [160]. Brofiga et al. also found that the hippocampal network mainly projects inhibitory connections to the cortical network using the hippocampus–cortex model in vitro [161]. In addition to the nervous system, Duc et al. co-cultured motor neurons and myoblasts on the neural chip platform and successfully activated the action potential of extracellular muscles by stimulating motor neurons with microelectrodes (Figure 8E) [61]. This study provides an effective model of the neuromuscular system, which can be used to study the underlying mechanism of physiological effects.

**Figure 8 micromachines-14-00709-f008:**
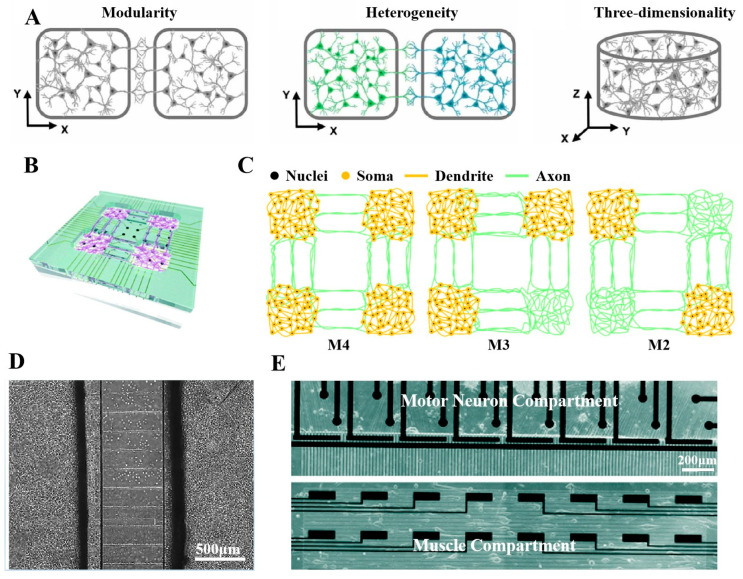
The application of neural chip platforms integrated with microfluidics and microelectrode arrays in the study of simplified brain models. (**A**) An ideal simplified model of the brain in vitro should have the following properties: modularity, co-culturing of heterogeneous cells, and three-dimensional cell cultures [98]. Reproduced with permission from Brofiga, Journal of Neural Engineering; published by IOP Publishing, 2021. (**B**) Neural chip platform for modular neuron culturing [27]. Reproduced with permission from Park, Lab on a Chip; published by Royal Society of Chemistry, 2019. (**C**) Three modes of co-culturing cortical neurons on a neural chip with four culture chambers. (**D**) In vitro brain model of hippocampal–cortex co-cultures [160]. Reproduced with permission from Chang, Frontiers in Neuroscience; published by Frontiers, 2022 (from Chang et al., 2022). (**E**) In vitro brain model co-cultured with motor neurons and muscle cells [61]. Reproduced with permission from Duc, Lab on a Chip; published by Royal Society of Chemistry, 2019.

In addition to modularity and heterogeneity, brain tissue also has a three-dimensional structure (Figure 8A) [98]. The three-dimensional microenvironment in the brain also plays an important role in the correct expression of nerve cells and normal electrophysiological activities. Therefore, when building a simplified model of the brain in vitro, the ability of the neuronal networks in the model to maintain 3D growth is also an important condition to evaluate whether the function of the model is complete. Brofiga and his colleagues built a three-dimensional interconnected heterogeneous (cortex–hippocampus) brain model on a neural platform integrating a microfluidic device and an MEA using microspheres as scaffolds [162].

Finally, we used Table 1 to summarize the typical applications of neural chips in this review article.

## 6. Conclusions and Future Perspectives

Neural chip platforms integrating microfluidic devices and microelectrode arrays have developed rapidly over the past decade. Compartmentalized microfluidic devices facilitate the control of the morphology of the neural network as well as the microbiochemical environment. Meanwhile, microelectrode arrays can feedback the changes in the neuronal network activity by detecting the electrophysiological signals networks. Overall, neural chips integrating microfluidic devices and microelectrode arrays have been developed into highly promising platforms for practical use, e.g., in vitro drug screening, mechanistic studies of neurological diseases, simplified model studies of the brain, etc. The neural chip platform has shown excellent application potential in neural correlation, and researchers have shown increased interest. Therefore, we comprehensively reviewed and focused on recent advances in the application of neural chip platforms.

Despite the exciting progress of neural chip platforms in neural fields, there are still many challenges. Below, we summarize several of the current major pitfalls of neural chip platforms. First, the space and maintenance of cell growth substances for cultures in microchambers and microchannels are greatly reduced compared to cultures exposed directly to air. This challenges the ability of cells to maintain long-term good activity [165] and greatly limits research of the neural chip platform. Second, there are some gaps between the functions of neuronal networks cultivated on the neural chip platform and those of brain networks [166]. This is not conducive to the development of in vitro models with more a complete brain function and organoid chips. Finally, there is still room for improvement of the fabrication of neural chip platforms. For example, the stability of the adherence of microfluidic devices to an MEA under long-term in vitro culturing can be improved. In addition, the application of more advanced interface materials can not only increase the biocompatibility of the chip platform but can also improve the recording performance and electrical stimulation performance.

In the future, researchers can make efforts in many directions to further optimize neural chip platforms integrating microfluidic devices and microelectrode arrays. First, the long-term cell culture technology in microfluidic devices can be further improved. For example, an automatic delivery system adding culture medium can be included in microfluidic devices, turning the cells in the microchannel into rich nutrients. In addition, improvements to the spatial structure of microfluidic devices can be an effective method. In studies of microelectrode arrays integrated microfluidic devices, the culture cycle of neural cells is usually less than two months [27,61,154]. To achieve some special experimental purposes, such as achieving the long-term tracking and detection of neural signals, to study the mechanisms of neuronal network growth and development, and to evaluate the long-term effects of drugs on neuronal networks, it is necessary for cells to maintain good activity in vitro for a long time. Secondly, to construct in vitro networks with structures and functions similar to those of neuronal networks in the brain, neural chip platforms integrating microelectrode arrays and microfluidic devices can achieve this goal using technologies such as 3D scaffolds. Third, advanced MEAs or microfluidic devices can be fabricated using new materials to obtain neural chips with a higher performance. For example, PEDOT and ITO are used as conductive layers for MEA to enhance the transparency of the chip [167]. This is helpful for observing the growth of neural cells in experiments. Nanohydrogel was used to wrap nanocomposites to improve the biocompatibility of microelectrodes [168]. We believe that with more research focused on solving such existing problems, neural chip platforms will make a significant contribution to neuroscience research.

## Figures and Tables

**Figure 1 micromachines-14-00709-f001:**
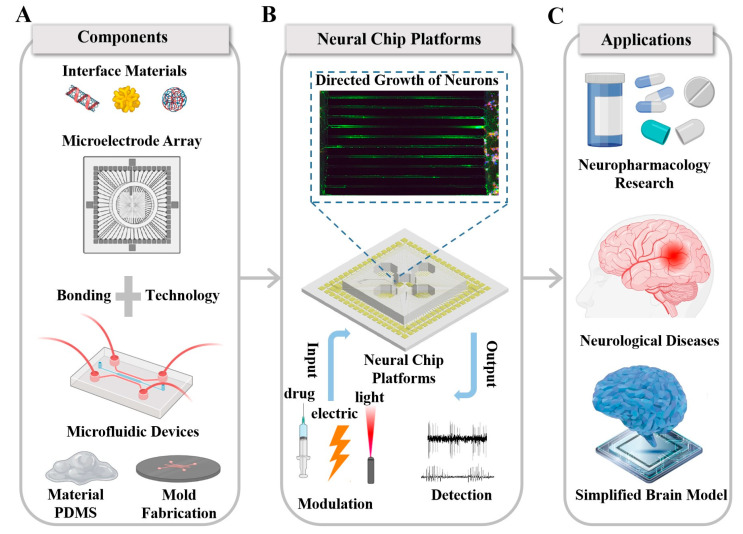
The framework of this article. (**A**) Microelectrode array and PDMS-based microfluidic devices. Materials and fabrication methods of microelectrode array and PDMS-based microfluidic devices. (**B**) The structure and usage of neural chip platforms (the fluorescence-staining diagram cited from Lewandowska et al. [29]) (**C**) Application of neural chip platforms, including neuropharmacology research, neurological diseases, and simplified brain models. (Created with http://BioRender.com (accessed on 10 January 2023)).

**Figure 3 micromachines-14-00709-f003:**
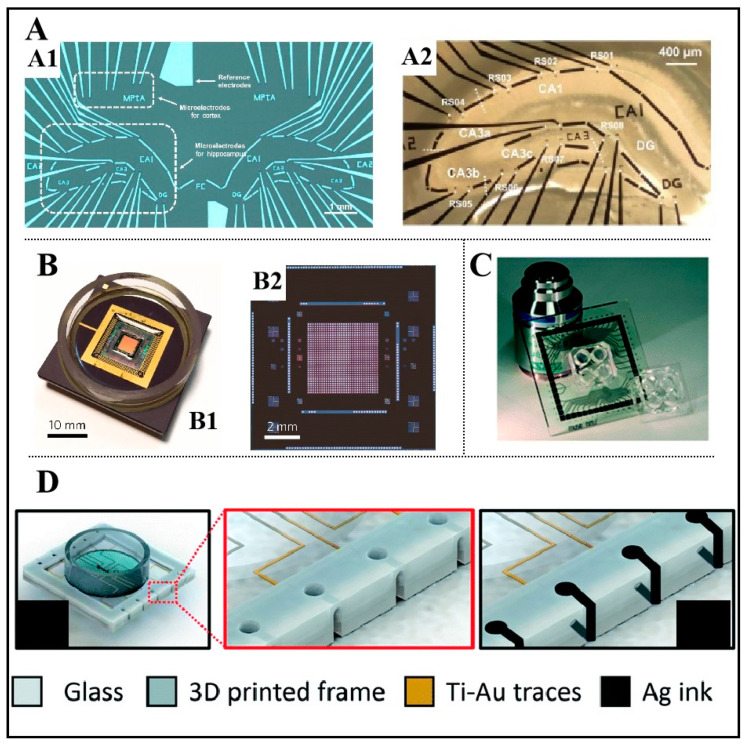
Future development direction of in vitro MEAs. (**A**) The internal structure (**A1**) and the real object (**A2**) of an MEA dedicated to rat hippocampal slices [57]. Reproduced with permission from He, Sensors and Actuators B: Chemical; published by Elsevier, 2021. (**B**) The object (**B1**) and internal structure (**B2**) of a high-density CMOS microelectrode array [60]. Reproduced with permission from Abbott, Nature Nanotechnology; published by Nature, 2017. (**C**) A multifunctional MEA combined with microfluidic technology [61]. Reproduced with permission from Duc, Lab on a Chip; published by Royal Society of Chemistry, 2021. (**D**) The use of 3D printing technology to simplify the process of making MEAs [65]. Reproduced with permission from Morales-Carvajal, RSC Advances; published by Royal Society of Chemistry, 2020.

**Figure 6 micromachines-14-00709-f006:**
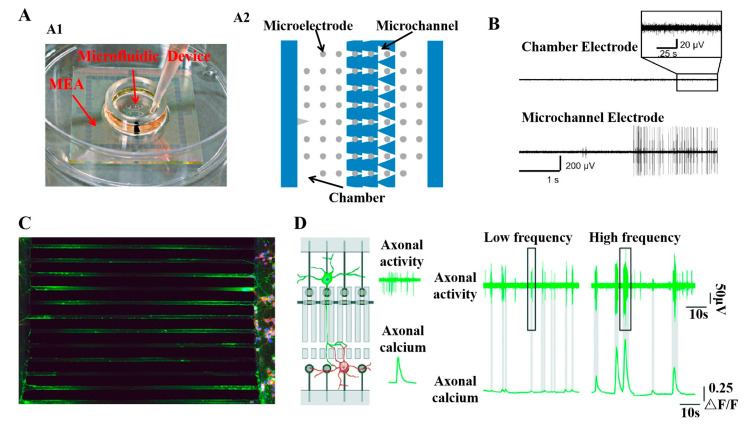
Neural chip platforms integrated with microfluidics and microelectrode arrays for recording neural information. (**A**) The microfluidic device and an MEA combination (**A1**), and the distribution of microelectrodes in microfluidic devices (**A2**) [133]. Reproduced with permission from Pigareva, Brain Sciences; published by MDPI, 2021. (**B**) The difference between the neuronal discharges recorded in the microfluidic device chamber (scale bars of electrophysiological signals: 20 μV, 25 s) and the microfluidic channel (scale bars of electrophysiological signals: 200 μV,1 s) [137]. Reproduced with permission from Goshi, Lab on a Chip; published by Royal Society of Chemistry, 2022. (**C**) The neurites in microfluidic devices characterized by immunofluorescence [29]. Reproduced with permission from Lewandowska, PLOS ONE; published by PLOS, 2015. (**D**) The neural activities of neurons in the microfluidic devices characterized by electrophysiological signals (scale bars: 50 μV, 10 s) and calcium signals simultaneously (scale bars: 0.25ΔF/F, 10 s) [143]. Reproduced with permission from Moutaux, Lab on a Chip; published by Royal Society of Chemistry, 2018.

**Figure 7 micromachines-14-00709-f007:**
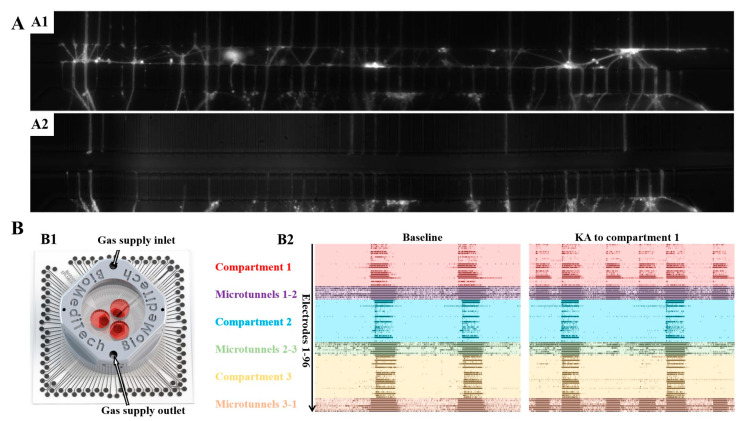
Application of neural chip platform-integrated microfluidics and microelectrode Arrays in neurological diseases. (**A**) Fluorescent images before and after axonal injury and the growth of axons before and after axotomy, (**A1**) and (**A2**) [151]. Reproduced with permission from Van de Wijdeven, Biosensors and Bioelectronics; published by Elsevier, 2019. (**B**) Building of an in vitro epilepsy model using KA to induce electrophysiological signals (**B2**) of neurons before and after epileptiform discharges on the neural chip (**B1**) [154]. Reproduced with permission from Pelkonen, Biosensors and Bioelectronics; published by Elsevier, 2020.

**Table 1 micromachines-14-00709-t001:** Several typical neural chips and their applications.

Neural Chip Components	Arrangement of MEA	Detection Object	Application	Reference
MEA	Microelectrodes fitting the shape of the dentate gyrus of the hippocampal slices	Hippocampal slices	Research on epilepsy circuit	He et al. (2021) [57]
MEA	128 microelectrodes are distributed in the center of the MEA	Hippocampal neurons	Research on the learning function of neural networks in vitro	Xu et al. (2022) [139]
CMOS MEA	4225 recording electrodes and 1024 stimulation electrodes	Retina	Research on visual restoration	Cojocaru et al. (2022) [163]
MEA and PDMS-based microfluidic device	Microelectrodes are in the microchannel	Neural stem cells	Detection of neurite signals	Kim et al. (2022) [141]
MEA and PDMS-based microfluidic device	Microelectrodes are arranged at the edges of three chambers	Human stem cell-derived neurons	Research on epileptic seizures	Pelkonen et al. (2020) [154]
MEA and PDMS-based microfluidic device	Microelectrodes are arranged on both sides of the microchannel	Co-culture of motor neurons and muscle cells	Construction of the neuron–muscle model in vitro	Duc et al. (2021) [61]
CMOS MEA and PDMS-based microfluidic device	26,400 electrodes located in an area of 3.85 × 2.10 mm^2^	Cortical neurons	High-density detection of neural signals	Duru et al. (2022) [164]
MEA, PDMS-based microfluidic device, and magnetic bead	60 microelectrodes are evenly distributed in two chambers of microfluidic devices	Cortical and hippocampal neurons	Cultivate three-dimensional brain network	Brofiga et al. (2020) [162]

## Data Availability

Not applicable.

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
