# Peer review of "Recent Progress and Perspectives on Neural Chip Platforms Integrating PDMS-Based Microfluidic Devices and Microelectrode Arrays"

_micromachines, 2023, doi:10.3390/mi14040709_

Round 1

Reviewer 1 Report

This review entitled "Recent Progress and Perspectives of Neural Chip Platforms Integrating Microfluidic Devices and Microelectrode Arrays" is a timely presented overview of important developments in the field of neural chip development. While the authors made a great effort to cover the research summarizing most popular device designs and application strategies, there is room for improvements of the paper.

Generally, it is advised to consider shortening of the review to pinpoint on relevant novel developments for Neural Chip Platforms rather than general microfluidic chip fabrication techniques applicable in many applications but not specific to neural chip platforms.

One other specific suggestion before going into the paper line by line is to reconsider the use of the wording "ex vivo" - generally, ex vivo refers to any investigated tissue investigated outside of the body (e.g. in the context here, for example, brain slices) but generally not to cultures that are made based on individual cells that reconstruct tissue or in the special case here "neural networks". Such culture formats are generally referred to as "in vitro", please check your wording and revise the document appropriately.  

Further revisions should be considered for the following suggested items, given here for the lines appearing in the document:

Suggested revisions:

Introduction:

L75: Generally culture chambers are not at at micron or sub-micro scale, i.e. references by [27]; however if this is specifically the case in the references given please make explicit in the sentence.

L86: "The phrase "neural science is introduced in blocks" is unclear there is also no reference to "blocks" in the later text. Please revise.

L92: Provide references for the Figures in the assembled Figure 1.

2. An Important Tool for the Regulation and Recording...

L117: "safe injection limit", do you mean "safe charge injection limit"? Please adapt accordingly to clarify.

L133: Figure 2 "H. Schematic diagram of printing 3D MEA with ink [45]." All other panels A-G in Figure 2 show the geometrical/morphological feature of the microelectrode. In panel H, this is not very clearly addressed. It is more a suggested fabrication method rather a material configuration being useful to microelectrode. Consider another image to show how the electrode with respect to material configuration looks like, or make more explicit how this printed 3D MEA electrode serve here for a feature dedicated to electrode functioning.

L139 it is suggested that here that nano-scale metal films are generally poorly performing on mechanical stability, biocompatibility and other defects. This is a statement that should not be made in general. Please revise and relate to specific cases where this has been demonstrated and cite the according literature.

L146: high temporal and spatial detection of neural information is not self-evident a property of CNT-based materials but are often also related to the layout/geometry of the electrode/electrode array; please make sure you explain in which sense CNT materials are better in this respect than other materials and why?

L153: the statement on stability and other properties listed in this context (L155 -157, 160) with respect to nanocomposites is to general. Explain based on the findings of the reference why this is an intrinsic property of the suggested materials.

L163 what does "multi-partition" mean in this context? 

L182: make sure referenced authors are cited correctly, i.e.  "Zips et al."  i.e. use capital latter to avoid confusion.

L190 what does the terminology "cross cutting" means in this context? Do the authors mean "multidisciplinary"?

L190-203 "We think that the future..." Reword the paragraph. This listing of findings form literature is not what the authors of this review think, it is actually depicting a trend in this literature that the authors of this review describe. Also with regard to the suggestion on the use of  "microLED" it is unclear from which references this statement is based on. If there are no references than the authors should retain from such loos statement. 

3. "An Important Tool for Neural Network Customization: ..."

L213-217 here several general statements are made, the authors should support these by references; on the other hand this paragraph should be an introduction to the various aspects being reviewed in this section; given the following subsections 3.1-3.3, this section introduction should be revised to better reflect on the content in the actual sub-sections.

In section 3.1 further it is recommended not to generalize the use of "polymer" as the main material in microfluidics; many interesting and valuable examples of microfluidic devices were actually made in glass and silicon. The authors should be more specific why PDMS has been so valuable in neural networks customization; or maybe this subsection is actually the more general introduction for PDMS chips in cell culture. Please clarify!

Section 3.2:

L248: Taylor et al. not "Taylar et al."

L256: Figure 4 C. "combinable"; please check wording. Also what is the device offering in terms of function? Also E. Does not related to function of the feature of the so called "stomach shape" ... while the other examples do relate to the function of the chip/feature of the chip. Please update.

L259: "the design of the mcirofluidic device"; to which specific device it is referred to here, else use the plural form i.e. "design of microfluidic devices"...

L261: just increasing number of chambers is not an indicator for "improvement", please revise.

L267 "can be spliced"; better maybe say "assembled" i.e. it is not clear what splicing in this context of work means.

L277 "designed a microchannel of "arch"...". Unclear wording; it is not possible what is mean here specifically with respect to requirements in design.

L281 What are "septal neurons"? Unclear. Please explain.

L283: how does the specific shape improved the efficiency of neurite growth? Do the authors mean that this specific "stomach-like" shape used by Forro et al. can effectively unidirectionally connect to reservoirs of cultures (i.e. if actually "node" does refer to a culture reservoir). Please clarify.

L285-286: ill-defined sentence. I.e. do the authors mean: "Girardin et al. [78] and Ihle et al. [79] studied the response of neural networks in chip devices inspired by the design of Forro et al. [30]? Please verify.

section 3.3:

L289: "the use of the wording "in the field of nerves" is not clear; do the authors mean "brain tissue models"? 

L292-293: cite according references for this statement on pathological mechanisms.

L295: "is usually constructed"; please do not generalize; just explain how the drug induced pathological model in vitro that the authors review are actually constructed and then just cite them. Apparently Kunze et al [80] is one and Kajtez et al [81] have constructed another example of such drug induced models. Followed by a few more examples.

L310: here the concept of co-cultures is introduced without specific references. Please add according citations.

L312: here it is stated that co-cultures enable a "simplified model"; this is not self-evident or obvious. Please explain along the citations selected.

L315: "...researchers build models."; Please clarify what models.

L320: How does the chip supports that the platform provides different brain regions? This is not self-evident or obvious. Please explain why.

L321: "..study the reward..." this is not a clear sentence; do the authors mean that the platform was used to study the neuronal circuit of the reward system by measuring synaptic activity?

4. Fabrication of Neural Chips...

Section 4.1: Clarify "ex-vivo" since many examples given here are actually "in vitro", i.e. cultured cells on MEA. It sounds like that the authors try to distinguish MEA development for implantable neural interfaces form these MEA-based neural platforms used in culture. But still then, it is believed that ex-vivo is not the proper term to use in this context of the examples given.

Overall, this section 4.1 does not review (cite references) where it should when listing the various features of MEAs. There are some references given in the Figure caption of Figure 5, but these are not clarified in the main text. Context unclear. Please revise.

L354: "submicron synapses"; synapses are by definition molecular entities, the addition of the word "submicron" makes not much sense here, else please explain carefully.

Section 4.2 ok in principle, however could be tailored to the specific topic of the section and potentially shortened significantly as the information given does not necessarily related to tech combined specifically with MEAs (at least not evidenced in the text itself, maybe the reference do provide information related to MEA?). Please revise accordingly.

Section 5.3 

L547: "The borrowed the method..." Do the authors mean "They used the method of Tong et al...." And please clarify who is "they", i.e. is this Wijdegen's team?

L558: "..etc, this neural chip platform...."; "this" refers to a specific one, but it is not clear which one. Probaly the authors could better use the plural form, i.e. :  "etc., neural chip platforms with integrated microfluidic devices also play an important role [134]."

L569: Figure 7 states "Neurochip Platform" like it is a brand name. If it is a brand name by a company, please state the supplying company name.

L 574: Section 5.4 "Brain simplified Models" unclear. Do the authors mean "The study of brain by means of simplified models"?

L592: what is meant with "they inoculated..." ?

L595: unclear what M4, M3 and M2 refer to. Please specify.

L596-600 this sentences are not clear. What does symmetric transition mean, and what are the "modular connections" connecting,  Wherefrom is it known that for connections of brain regions "specific module interconnection models" are more suitable? Please add appropriate citations. And, what is a "module interconnection model" actually? Please clarify the wording.

L603: "in Poli's work" is Poli the sole author of this work? And, to which specific work this statement relates to? Please provide citation probably. Now, the reader could assume it is [140]  but it is not evident form the way it is described by the authors of this review paper.

L606-607: What design feature of the chip makes "the circuit" cultured on the chip reminiscent for hippocampus tissue?, when culturing form individual cells takes place. Or is the culture performed as a slice or in a particular scaffold?

L609-610 Unclear which neural chip platform is used. Does the statements of L609-612 refer to the citations [142], [82], [143] and [144]? Please clarify.

L624: .."to study inner physiological effects." What is meant with "inner"; please clarify.

L643: "..three-dimensional structure (Figure 8A)." Needs a references here in the main text.

L638 "rational" or "relevant" what is meant here? Please clarify.

L641-643: The information provided in this lines is not well embedded in the context of the remainder of the text of this section. 

6. Conclusions and ...

L648-649 "...with MEA makes the small change of neural network...." sentence is not well constructed and hence unclear.

L655: "we propose"... this is the conclusions section, hence the review is performed already rather than proposed to be done.  Please reword.

L659-660: "We attempt.." this sentence is superficial in line with what is already said before. Please remove.

L665-666: It is a bit to dogmatic to suggest that these is the main reason, as actually now references have been cited that review this context of challenge with regard to "maintaining better activity for a long time".

L668" The topic of maturation was not actually discussed in this review. Either add according references in your review text in the main sections of this review or remove the statement here form the conclusions.

L673: light transmission and other functions as a function of interface materials has also not been discussed in the main sections of the review. Please either support in the main text such research or remove form the conclusion.

L677: With regard to long term culture: actually the main text in the sections above never discussed any days in culture for the many examples of microfluidic integrated MEA chips in this review. Please clarify on this topic; then state here more precise what is meant with long-term cell culture technology and why this is important being considered as a perspective.

L680-681 "...innovative improvement ...is also an effective improvement.." superficial statement/trivial; as the complete design process of microfluidic chips is generally application driven.

L 681-682:  It is not reviewed and not self-evident that 3D scaffolds can be used to build a multi-functional neural network. What do the authors of this review consider being "multi-functional" with regard to neural networks presented in a chip culture format? Since this topic is not reviewed in the main text of the sections above, apparently this is a perspective topic, then please be more explicit in what from the above reviewed material gives you as authors of this review the confidence that 3D scaffolds will add to multi-functionality.

L683: "..a variety of functional materials..." Since functional materials are not defined in this review specifically, it is unclear what kind of properties arising from the material itself can help to gain the suggested performances or improvements for neural culture. Please reconsider such general statement and at least make sure to the reader through the referenced material in the main part of this review that a reader can follow the authors in what these materials are as class or classes of materials.         

Author Response

Please see the attachment.  The revisions were highlighted (using the “Track Changes” function in Microsoft Word) in the revised_manuscript.

Reviewer 2 Report

This review first introduces the latest technological trends in microelectrode arrays and microfluidic devices each. Then, it describes the fabrication processes and applications of neural chip platforms integrating the MEA and the microfluidic device. Overall, the review provides an adequate overview of previous works.

However, before discussing the detailed techniques for MEA development in Chapters 2.2 – 2.3, I suggest explaining the general composition of the MEA first. A description of the MEA structure cannot be found until Chapter 4. It would be helpful to understand the research on MEA performance improvement described later.

Other minor comments are listed below.

- Although the review does not cover organoid research, it is mentioned in the abstract (Line 27).

- The article ‘an’ or ‘the’ is required before ‘MEA’.

- In Chapter 4.2, the text explains PDMS device fabrication first and mold fabrication next, but Figure 5B illustrates in the opposite order, so it is a bit confusing.

- In the paragraph introducing ‘Heterogeneity’ (from Line 609), it would be better to refer to Figure 8A once more.

- In Figure 8E, there is a typo: ‘Motor Nuron Compartment’. Also, check for other typos (e.g., uppercase ‘D’ on Lines 129 and 217).

- Check the Bibliography (e.g., capital letters or abbreviations)

Author Response

(The authors gave the same response as above.)

Reviewer 3 Report

Title: Recent Progress and Perspectives of Neural Chip Platforms Integrating Microfluidic Devices and Microelectrode Arrays

This manuscript provides a review of neural chip platforms integrating microfluidic devices and microelectrode arrays. The review presents recent progress in ex-vivo microelectrode arrays (MEA) and PDMS-based microfluidic devices. The topic of the manuscript is a promising field that is relevant to a wide audience. Overall, the manuscript is well-organized and easy to follow. However, there are some issues that need to be addressed. The specific comments are as follows:

1.     In section 2, the authors mention the recording and regulation of neural information several times. However, there is no clear definition of these terms. Additionally, the illustration of regulation in Figure 1 is not clear.

2.     Figure 1 provides a great illustration of the overview. However, the authors should provide a more detailed caption to explain the three parts of the figure.

3.     In section 2, the authors mention ex-vivo MEA and in vitro MEA. They should justify the difference between these and modify the expression to make it clear.

4.     In section 2, the authors introduce the materials and development of ex-vivo MEA. However, they do not clearly explain how MEA works for the neural chip or why different types of MEA are designed to improve the recording and regulation of neural cells. In other words, the mechanism of MEA acting on neural cells should be better addressed.

5.     There are different types of microfluidic devices, but this manuscript only introduces PDMS-based microfluidics. Therefore, it is suggested to change the title from "microfluidic devices" to "PDMS-based microfluidics" to specify the scope.

6.     The authors should consider adding a table to show the different types of neural chips, the geometries of the microelectrode array, and their applications in the neural field. This would provide a better understanding of the different platforms and their capabilities.

Author Response

(The authors gave the same response as above.)

Round 2

Reviewer 1 Report

The article should just go through a final English check for minor spelling mistakes, but is contentwise ready for publishing. It is still somewhat lengthily. editor and Authors could probably work together to make the manuscript slightly more during production phase.